# Results of Nurse Case Management in Primary Heath Care: Bibliographic Review

**DOI:** 10.3390/ijerph17249541

**Published:** 2020-12-20

**Authors:** Vicente Doménech-Briz, Rosario Gómez Romero, Isabel de Miguel-Montoya, Raúl Juárez-Vela, José Ramón Martínez-Riera, María Isabel Mármol-López, María Virtudes Verdeguer-Gómez, Álvaro Sánchez-Rodríguez, Vicente Gea-Caballero

**Affiliations:** 1Hospital Universitario de la Ribera, km 1, Ctra. Corbera, 46600 Alzira, Spain; vidobri@hotmail.com (V.D.-B.); alvarovalencia97@hotmail.com (Á.S.-R.); 2Nursing School La Fe, adscript center of Universidad de Valencia, 46026 Valencia, Spain; demiguel_isa@gva.es (I.d.M.-M.); marmol_isa@gva.es (M.I.M.-L.); gea_vic@gva.es (V.G.-C.); 3Research Group GREIACC, Health Research Institute La Fe, Avda. Fernando Abril Martorell, 106. Pabellón docente Torre H, Hospital La Fe, 46016 Valencia, Spain; 4Department of Nursing, University of La Rioja, 26006 Logroño, La Rioja, Spain; 5Research Group BMP Idi-Paz. Hospital La- Paz, Paseo de la Castellana, 261, 28046 Madrid, Spain; 6Departamento Enfermería Comunitaria, Medicina Preventiva y Salud Pública e Historia de la Ciencia, Universidad de Alicante, E-03080 Alicante, Spain; jr.martinez@ua.es; 7Dirección Atención Primaria, Departamento de Salud de Alzira, Alzira, Valencia, Spain. km 1, Ctra. Corbera, 46600 Alzira, Valencia; mariviverdeguer@gmail.com

**Keywords:** nursing, chronic disease, case management, patient outcome assessment

## Abstract

Background: The new characteristics of today’s population, together with the presence of chronic diseases in the elderly, require a new approach to care, promoting coordination between different levels of care. In this sense, we find the figure of the nurse case manager (NCM) in primary health care mainly responsible for ensuring continuity of care in complex patients with chronic diseases. Objective: to describe the role of the NCM in care management, determining its effectiveness in addressing chronic disease (health outcomes and quality of life) and its efficiency in the health system. Methods: Bibliographic review of scientific evidence on case management applied to nursing. Between March and April 2020 a bibliographic search was carried out in the Dialnet, Scielo, Scopus and Pubmed databases. Inclusion criteria: articles written in the last 5 years, which analyze how this nursing rol influences the care and health of patients. Results: A total of 16 articles were selected. The NCM reduced the use of the emergency department, hospital admissions, readmissions, and the duration of these in the patients studied. Conclusion: The NCM is effective and efficient for both patients and health institutions, and a common practice model is needed that includes standardized protocols and evidence-based practices.

## 1. Introduction

In all health services, placing the patient at the center of the care system is linked to the quality of the health system. Thus, the user ceases to be a passive subject who receives information and becomes an active user, who has more information about his or her health and rights and has a more proactive attitude [1]. In this health care, nurses are key professionals, because scientific evidence indicates that the lack of nurses generates a situation of significant risk for the health of patients [2].

The health status of the population has generally improved in recent years; however, “long-term disability and chronic diseases are increasing due to a rapidly aging population” [3]. In this line, attention to chronicity is a challenge for any health system, also for Spain, due to the consequences of the aging population, the need to respond to the increase of users with chronic diseases and the high existing comorbidity [4]. The attention to all these health problems entails a high cost in services and benefits, which must be faced. This situation requires the implementation of new professional nursing profiles, with expanded and advanced skills that are developed in response to new demands in different contexts and areas of care. The figure of the Nurse Case Manager (NCM) is a good example of an emerging professional figure worldwide [4].

Case management was born in the United States (USA) in the years 1950/1960, applied for the first time in mental health cases; later it was used for the care of people with high-risk health problems and high costs, “with the aim of improving efficiency and decreasing variability” [5]. Case management is “the collaborative process by which the options and services needed to meet an individual’s health needs are assessed, planned, implemented, coordinated, monitored, and evaluated, articulating communication and available resources to promote quality and cost-effective outcomes” [5]. The NCM has become a common figure also in Spain, to offer care in a changing society, and to people with complex situations derived from chronicity, pluripathology, frailty and aging [4]. In Spain, the NCM plans and coordinates the care of patients with chronic diseases, mainly in the area of Primary Health Care (PHC), especially those more complex cases that require better coordination of the health system [4].

Despite the fact that case management has not been carried out exclusively and always by nursing professionals, nurses have usually been the figure chosen in different healthcare settings, although doctors or social workers have also carried out this work [6]. The fact that nurses usually lead case management is because they have a comprehensive vision of covering human needs, promoting self-care, as well as assessing and planning care. The NCM uses the nursing process as a dynamic method with a theoretical basis that allows for humanistic care, focused on optimally achieving objectives. This nurse has an in-depth knowledge of the bio-psycho-social context of the user (and his or her community, networks, environments etc.), which facilitates joint decision-making with other professionals when planning care [7], assuming responsibility for case management.

### Background

The case management model followed was the Chronic Care Model (CCM), developed by E. Wagner, in 1998 (USA) [8]. The CCM considers the following elements as a requirement for managing chronic diseases: the health system, clinical information systems, patient decision support, patient self-management, delivery system design, and the community (including patient organizations and resources) which interact with clinical practice [8]. The main objective of this model is for patients to be active and informed at all times during the clinical relationship, through proactive professionals with the skills to perform quality care and excellent health outcomes (along with a high degree of patient satisfaction) [8,9]. The ultimate idea of this model is that patients are the ones who will finally take care of their disease, thanks to the elements seen above.

Furthermore, the Kaiser Permanente risk stratification model, which classifies patients according to the level of care they need, has also been considered as a complementary model [7,10]. Thus, at the top of the pyramid there are the highly complex chronic patients, which require specific care directed by professionals with advanced levels of competence, such as the NCM, capable of having an exhaustive knowledge of the patient and his or her environment, and can provide multidisciplinary work and continuity of care [4,6,7,8].

The case management model is a way to improve the Integrated, Coordinated and Continuous Care, allowing to establish a link between the Hospital Care (HC) and the PHC. With this, the articulation of these care areas is improved, avoiding loss of care that affects the integrity and continuity of the health system [6]. With the work of the NCM these problems of the patient who needs health care in multiple environments are solved, which generates more complexity and fully justifies the need for case management.

In more current terms, it is pointed out that case management is a model of advanced nursing practice [11], which is holistic, patient-centered (in its environment), promotes personal autonomy and social participation, and facilitates access to resources. In addition, some of the tasks of the NCM are the recruitment of dependent patients who need a comprehensive multiprofessional approach; to identify people with higher pathological burden and high risk of hospital admission; to try to perform the least invasive care; to establish alliances between PHC and HC professionals; to ensure coordination and continuity of care in patients who require complex care and treatment at different levels of care; to make a personalized plan of care (including the PHC giver in this plan); and to promote individual and family self-care [4,6].

Since the type of patients with whom the NCM works are users with chronic diseases and multimorbidity, especially complex patients, it is interesting to analyze the results of an advanced practice figure such as the NCM. This is important because currently long-term disability and chronic diseases have increased, with a severe socio-demographic transformation as a result of the aging population that underpins the redesign of health services and the skills of their professionals [3].

The research question that emerges from the literature review is: How do the activities performed by nurse case managers affect the management of patient care? The patient is the person receiving care, the intervention is the activities performed by these advanced practice nurses, and the outcomes are those derived from care management and efficiency. Therefore, we intend with this study to describe the role of the NCM in care management, determining its effectiveness in addressing chronic disease (health outcomes) and its efficiency in the health system.

## 2. Methods

### 2.1. Study Design

Bibliographic review was carried out between January and April 2020.

### 2.2. Search Strategy

Two independent researchers consulted the following electronic databases: PUBMED, SCIELO, SCOPUS and DIALNET. Terms from the “Medical Subject Heading“ [12] (nurse, case management, crhonic disease) and the DeCS [13] (enfermera, enfermedad crónica, gestor de casos) were used. Additionally, the free term “care management” was used too. Subsequently, a manual review of the bibliographic references of the selected articles was carried out in order to include other studies potentially valid for the review through the reverse or secondary search method (Table 1). The last search was conducted on 30 April 2020.

### 2.3. Selection Criteria

Inclusion criteria: articles describing the activity of NCM or case management applied to nursing; articles related the NCM to chronic diseases; studies describing the efficiency of the interventions carried out by NCM (cost reduction, reduction of complications etc.).

Exclusion criteria: gray literature (web pages, conference abstracts, unpublished theses etc.); clinical practice guidelines; published articles before 2015 (in Scopus the filter of years 2016–2020 was used, due to the limitations of the database); articles describing the role of NCM without studying the results of their interventions.

### 2.4. Research Variables

The information obtained was grouped on two variables: effectiveness (hospital admissions, days of hospitalization, improvement in health status, satisfaction) and efficiency (use of resources and cost of care).

### 2.5. Methodological Quality and Level Of Evidence

To evaluate the quality of the studies obtained, we used the Critical Reading Methodology of the Spanish Critical Appraisal Skills Programme (CASPe, Oxford, UK) [14] and the quality criteria of the STROBE statement (Strengthening The Reporting of Observational Studies in Epidemiology) [15]. Besides, the degree of evidence and the level of recommendation through the Scottish Intercollegiate Guidelines Network (SIGN) scale were also assessed [16].

### 2.6. The Selection Process of the Studies

The selection process of the studies was carried out through the reading of the title and abstract of each article. After the elimination of duplicates and the application of the inclusion and exclusion criteria, the articles were read in full, with an evaluation of the quality of each article with critical reading guidelines. Finally, the references of the selected articles were reviewed in order to apply the reverse search method.

The result of the literature search process is shown in Figure 1.

## 3. Results

Sixteen articles were selected after the search procedure: 13 were obtained from the bibliographic search and 3 were included by reverse search. Among them were two descriptive studies [10,17] both published in Spain. López and Puente [17] describe the degree of institutionalization of the NCM, while Valverde et al. [10] analyzed the functions and target population of NCM. We also found six literature reviews. Of these, two bibliographic reviews [18,19], carried out in Spain and the USA, respectively) and four systematic reviews [20,21,22,23], the latter is a meta-analysis. Of these studies, one was carried out in the United States [22], two in the United Kingdom [20,23] and one in South Korea [21] (Figure 2). With regard to the longitudinal observational studies, two were obtained from follow-up cohorts served by case management services [24,25]). In addition, we found two experimental studies; a clinical trial [26] and a randomized clinical trial with a control group [27]. In both, it was observed how the activities of the NCM influence the care of patients. One pure qualitative study [28] was retrieved in which 23 nurse case managers in South Korea were interviewed in focus groups and individually. Finally, three studies of mixed methods design were obtained [29,30,31] (Figure 2).

### 3.1. Health Effects: Efficacy Results on Patients with Chronic Diseases

The interventions carried out by the NCM were more effective and efficient in addressing people with chronic diseases than those carried out according to the traditional model [28]. This nurse can provide the same quality of care as physicians in a wide variety of services (such as routine monitoring of chronic patients and first contact with patients with minor health problems). Improvement in disease self-management, knowledge of the disease and quality of life was observed in the patients studied [25,27].

Case management has proven effective in the treatment of many chronic diseases, such as diabetes, hypertension, obstructive pulmonary disease, and cancer care [18,26,27]. This was due to the interventions made by the nurse, which included the regular presence of the NCM with the patient and family in the consultations, home visits, telephone contacts (a higher level of care), aids in daily decisions for disease control (individual patient planning) and the structuring of a follow-up protocol for complications.

The interventions described above agreed with those analyzed by Joo and Huber [22] and Davisson and Swason [31], in which the objective of the NCM was to provide continuous care for the patient with optimal follow-up to improve his or her health condition. Caregivers were also supported through active listening and emotional and instrumental support so that they could continue to perform their tasks, providing the best possible care to patients.

The aspects most valued by participants concerning to the case management program were its accessibility, both by telephone and in person; always being cared for by the same professionals and the interpersonal relationship established between the nurse and the patient [31]. This professional-patient relationship made it possible for the nurse to advise the patient so that he or she would not immediately resort to self-medication (this new form of care helped patients to understand their pathologies and to detect signs of alarm, thus regaining the effectiveness of the self-management described above [25,27]).

The improvement of psychological outcomes was analyzed in studies that had several similarities, such as a gradual transition of care from hospitals to communities, continuously through follow-up by home visits or telephone calls (coordinated and continuous care from the hospital environment to the communities). These interventions provided comprehensive care, referral services, patient support, and the establishment of common and agreed-upon goals [10,21,22,30].

Considering the unfavorable results for case management models, a meta-analysis found that there are no significant differences in patient mortality [22]. However, very small significant effects were observed that favored case management for self-reported health status in the short term, for patient satisfaction (short and long term), and for functional health, as we had seen previously [10,18,19,21,24,29,30,31].

### 3.2. Profitability Results: Efficiency

The NCM reduced the use of the emergency department, hospital admissions, readmissions, and the duration of these in the patients studied [10,17,21,22,25,29]. In this way, there was a reduction in direct and indirect costs for the health system since, with less frequent use of the hospital emergency department, patients underwent fewer diagnostic tests and with the accessibility of the case management program, they required fewer hospitalizations [17,20,21].

Furthermore, there was greater activation of social resources when patients with chronic diseases are treated by the NCM [24,25,29,30]. Furthermore, it was described that patients with low income and lower educational levels achieved the same benefits as those observed in patients with higher income and higher levels of schooling (in terms of diabetes treatment), thus generating an efficient and effective cost to improve the complications of diabetes [26], reducing social inequalities.

The only study that did not show favorable results on the effectiveness and efficiency of case management was a meta-analysis [23], in which no significant differences were seen in the total cost or the use of primary or hospital care.

Table 2 summarizes the content of the documents selected in the literature search.

## 4. Discussion

With our study, we aimed to know the most effective interventions of the NCM to improve the health of the population, and the resulting benefits for the health system. Most of the studies consulted previously agreed that the interventions related to a case management program, generally developed by nurses, had a positive impact on people’s health and generated savings in health institutions [10,17,18,19,21,22,24,25,26,27,29,30,31].

The fact of being able to analyze articles from different countries allows us to understand that health data of people with chronic pathologies and comorbidities are quite common all over the world, predominantly elderly chronic patients who generate a high cost in health care [10,17,19,20,23,24,28]. For this reason, strategies have been implemented to generate savings by optimizing the skills of professionals, mainly nurses. The interventions carried out by the NCM have shown a comprehensive nature, assessing the needs and context of the person to provide care that responds to their health problems [10,17,18,19,21,26,27]. These activities are aimed at reducing the defragmentation of health services in order to provide coordinated and continuous care [10,19,23,31].

### 4.1. Health Results: Effectiveness

The interventions carried out by the NCM were comprehensive, assessing in depth the needs and context of the person to provide care that responds to their health problems [10,19,21,26,27]. These activities are aimed at reducing the defragmentation of health services to provide coordinated and continuous care [10,19,23,31]. Besides, we found a high rate of patient satisfaction with the services offered by the NCM, because this nurse spends more time with patients, giving them more information and advice, especially in chronic follow-up processes [18,19,26,31]. With the exception of the last referenced article [31], it is important to note that patient satisfaction with the role of NCM is supported by an adequate level of evidence.

In the literature consulted, it was observed that nurse case managers improved certain health outcomes: reduction of high blood pressure by facilitating adherence to medication [19,24,27]; reduction of glycated hemoglobin in patients with type II diabetes mellitus due to this nurse’s contact with the patients [19,26]; and decrease the consumption of addictive substances by offering users the services available from the health system by following up with patients who have more frequent contact with them [19,22]. The high level of evidence from these studies gives us confidence that improving clinical outcomes in patients is a fact that reinforces the need for NCM. All these studies also highlighted the empowerment achieved by both the patients themselves and their caregivers, thanks to the training that the NCM offered, improving coping with chronic diseases (high blood pressure, diabetes, chronic obstructive pulmonary disease or use of addictive substances) [18,26,27,31]. Some of the activities carried out by the NCM were the regular presence of this professional with the patient and his or her family in the consultations, help in the daily decisions, structuring a follow-up protocol for complications, support for caregivers or telephone follow-up [22,25,26,27,31].

Several studies affirmed that the NCM is capable of producing better results in patients with mental illnesses [20,22,30], thanks to motivational interviews or the greater contact of this Advanced Practice Nurse, with advanced competencies, with the patients. Although there is also literature that does not favor case management, the same authors show improvements in terms of patient satisfaction or functional health [23].

### 4.2. Profitability Results: *Efficiency*

As a result of the case management program, the above-mentioned studies indicate that there is a decrease in the use of health services related to complications of patients’ pathologies, as well as a lower risk of complications such as between hospitalizations, emergencies or readmissions. Due to this, there were evident savings in health costs, which shows us how profitable the services offered by the NCM are [19,21,22]. Although the details of the case management programs varied among the studies analyzed, all mention a continuum of follow-up care through home visits or phone calls made by the NCM. The authors described that regular contact with the case manager may have influenced the results of hospital utilization [10,17,18,19,21,22,24,25,26,27,29,30,31]. Although the efficiency of NCM appears to be demonstrated, it is true that the evidence may be controversial as a result of the type of studies implemented.

Some aspects well-valued by patients were the accessibility (both telephone and face-to-face), the continuity of care (same nurse) and the interpersonal relationship established between nurse and patient, in addition to the perception of greater support by health professionals. These benefits have been concluded by various studies [21,25,26,27,31], which affirm that the NCM makes it easier for clients to know more about their illness and to avoid having to go to the emergency services in case of doubts or complications. The evidence found seems to indicate that, indeed, the accessibility and type of patient relationship with NCM is a strength for the health system.

The only study that does not support the case management model is the one conducted by Stokes et al. [23], who argued that this type of care is not effective for those patients called “at risk” (co-morbid) in PHC. This study only describes some improvements in patient satisfaction. These conclusions may be debatable, since Davisson and Swanson [31] defend that the patients themselves recognized the nurse as the most important element of the program, being the professional most linked to the care and attention of the patients. Even so, the fact that most of the literature consulted expresses the improvement of the health of the patients studied after the implementation of a case management program highlights the importance of NCM as a key figure for health systems.

Most studies [18,20,21,22,23,28], all of them with strong or acceptable evidence, show that nurses did not engage in evidence-based practice due to the lack of a model integrating NCM interventions. As a result of this problem, in some reviews [20,21,24] a great variety in the interventions carried out has been observed, which leads to the fact that, although effective and efficient results are generated, they cannot be generalized and analyzed in greater depth because there is no common theoretical model that uniformly encompasses all the practices of these nurses.

### 4.3. Limitations

Some of the limitations of this literature review are that the NCM activities in various countries have been contemplated, so that, despite the fact that the activities carried out are similar, the socio-cultural context and the level of academic development and competence of nurses in each country may influence them. Furthermore, by considering different chronic diseases, it has not been possible to make a more comprehensive comparison between case management intervention programs applied to a single disease.

On the other hand, some studies did not expressly talk about the NCM, but stated a model of case management in general (while it is true that most of these models were made up of nurses, some also included other professionals, such as social workers). It would therefore be necessary to carry out new research focused on the model developed by nurses, to describe the effects obtained without generating possible biases.

The new research should measure the time from which the NCM begins to generate positive effects for the health of patients, an aspect that has not been studied. In addition, they should incorporate standardized protocols/common clinical practice guidelines and a proprietary model of care, to avoid the variability of interventions found during the literature review.

## 5. Conclusions

The interventions carried out by the NCM have an integral character, considering the objectives and needs of each person, and providing coordinated and continuous care. The activities carried out by the NCM improve the health of people with chronic diseases and comorbidity, allowing better results in health indicators. Case management is effective and efficient in most of the literature consulted, reducing health care costs and improving the quality of care.

## Figures and Tables

**Figure 1 ijerph-17-09541-f001:**
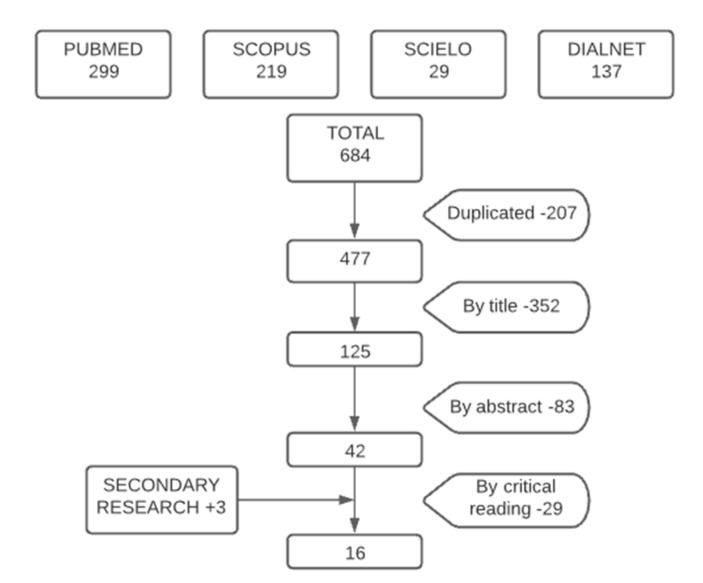
Flow chart of research.

**Figure 2 ijerph-17-09541-f002:**
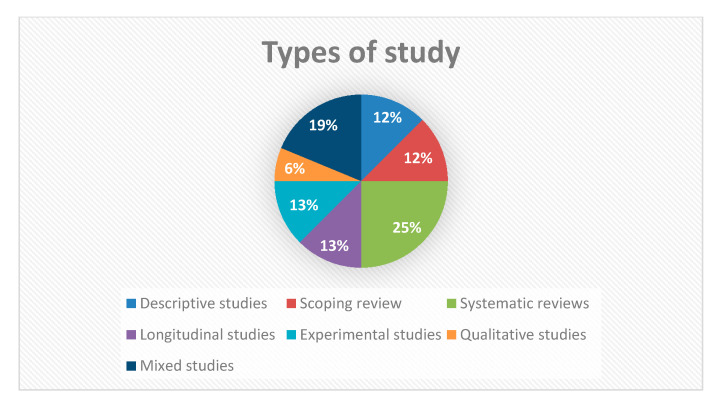
Types of study designs.

**Table 1 ijerph-17-09541-t001:** Search strategy.

Data Base	Search Strings	Search Period	Obtained Articles	Selected Articles
PUBMED	(nurse) AND (case management) AND (chronic disease) AND (care management)	2015–2020	299	6
SCIELO	(nurse) AND (case manager) (nurse) AND (case management) (nurse) AND (case manager) AND (chronic disease) (case management) AND (nurse)	2015–2020	0 + 7 + 1 + 21 = 29	1
SCOPUS	(nurse) AND (case management) AND (chronic disease)	2016–2020	219	4
DIALNET	(nurse) AND (case management) (nurse) AND (case manager) AND (chronic disease)	2015–2020	131 + 6 = 137	2
SECONDARY SEARCH		2014 *–2020		3

*** As an exception to the inclusion criteria, an article published in 2014 was included in the results, as it was considered relevant to the objectives of the review [10].

**Table 2 ijerph-17-09541-t002:** Information summary table.

Authors/Year/Country	Design/Sample	Aims	Variables	Results	Conclusions	Quality, Level of Evidence and Recommendation.
Valverde Jiménez et al., 2014. Spain.	Descriptive and cross-sectional, observational study. ‑ 258 patients and 115 caregivers.	‑ Study NCM’s process maps ‑ A visit is made in consultation or at home for initial assessment.	Sociodemographic and clinical profile of the target population. Caregiver overload.	Since the incorporation of the NCM, there has been an increase in the recording of both patient and caregiver assessments by patterns.	‑ The collected population is aged, fragile, with a severe level of dependency, preferably female, with family caregivers (female) where those who are not overburdened are more frequent.	STROBE: 20/22 SIGN: 3 D
Castanho, de Fátima & Verdú, 2015. Brazil.	Clinical Trial. Sample: 80 patients, randomized sample.	Comparation nursing case management to usual care in the treatment of glycosylated hemoglobin in patients with type II diabetes mellitus.	Glycated hemoglobin at the beginning of the study, at 6 months and at 12 months.	‑ Glycated hemoglobin was reduced by an average of 9.0-10.33% in patients enrolled in nursing case management, and by 9.57-8.93% in the control group.	‑ The implementation of the case management model reduces the HbA1c levels of people with type II diabetes mellitus over time. ‑ The case management program provides greater patient contact and improving glycemic control, also reduces the risk of chronic complications. ‑ People with fewer resources can have better glycemic control thanks to the case management program.	CASPe: 7/11 SIGN: 1+ B
Stokes et al., 2015. United Kingdom.	Systematic review with meta-analysis.	To describe the effectiveness of the case management program in PHC risk patients.	‑ Perceived health status, mortality ‑ Total cost of health care ‑ Use of health care services ‑ Patient Satisfaction	‑ No significant differences were observed in the total cost of health care, mortality or use of PHC and Specialized Care. ‑ Perceived health status increases very little in patients with chaos management programs, as well as in patient satisfaction. ‑ The effectiveness of NCM can be increased when it is performed by an interdisciplinary team.	‑ The results do not support case management as an effective model, especially in terms of reducing health care use or total costs.	CASPe: 9/10 SIGN: 1+ A
Reilly et al., 2015. United Kingdom.	Systematic review (only randomized controlled trials were included)	To summarize the results of health indicators of the health system.	‑ Probability of being institutionalized ‑ Stay in hospital unit ‑ Number of people admitted to the hospital at six months ‑ Mortality. ‑ Quality of life ‑ Caregiver’s charge ‑ Total cost of services at 12 months.	‑ The group of patients included in the case management program was significantly less likely to be institutionalized at six months, just as there is a reduction in the number of days in a hospital unit in these patients at six and twelve months. ‑ There was no difference in the number of people admitted to the hospital, nor in quality of life and mortality.	‑ There is some evidence that the case management model is beneficial in improving some outcomes at certain times for both the person with dementia and their caregiver. ‑ There was great heterogeneity between the interventions and the outcomes measured.	CASPe: 9/10 SIGN: 1++ A
Joo & Liu, 2015. South Korea.	Systematic review (only randomized controlled trials were included).	To synthesize evidence of case management programs in patients with chronic diseases.	‑ Hospital use. ‑ Returns. ‑ Visits to out-of-hospital emergency services and hospital emergency units.	‑ Case management greatly reduces hospital readmissions and emergency room visits ‑ Recovered studies tended to report positive results.	‑ Case management helps reduce hospital use for patients with chronic diseases. ‑ Implementation of the nurse-led case management program, but with a multidisciplinary approach, results in significant decreases in readmissions, emergency department visits, hospitalization days and health care costs. ‑ Case management improves the hospital’s cost-effectiveness and the quality of care received.	CASPe: 10/10 SIGN: 1+ A
Joo & Huber, 2016.United States.	Systematic review (only randomized controlled trials were included).	To summarize evidence of a substance abuse case management program in community settings.	‑ Decreasing substance abuse ‑ Less social problems ‑ Lower utilization of health services ‑ Increasing patient satisfaction	‑ Reduction of substance use, family or social problems, in patients receiving case management. ‑ Reduction of utilization’s health service among patients receiving case management.	‑ The services offered by the case management program provide customer-centered, continuous, quality and cost-effective care.- Case management helps coordinate care. ‑ More interventions and studies are recommended to support case management on a practical level.	CASPe: 9/10 SIGN: 1++ A
Tortajada et al., 2017 Spain.	Longitudinal cohort follow-up study Sample: 714 patients with complex multi-morbidity.	To analyze a case management program (patient assessment, education regarding prescribed medication or lifestyle, patient follow-up)	‑ Promotion of self-care ‑ Treatment compliance ‑ Identification of risk situations ‑ Reduce unplanned income risk.	‑ The program reduces the number of unplanned admissions and emergency visits. ‑ Reduction of hospital stay. ‑ Accessibility to health care is facilitated ‑ Higher self-care capacity	‑ The case management program has produced significant savings in hospital resources. ‑ Positive impact of program interventions on chronic patients. ‑ The primary health care team along with the NCM and other professionals can address the needs of multi-morbid patients.	CASPe: 11/11 SIGN: 2++ B
López Vallejo & Puente Alcaraz, 2017. Spain.	Descriptive study ‑ Comparison between autonomous communities through a reference standard regarding the approach to chronicity.	‑ To compare between different communities the approach to chronicity by institutionalizing the NCM in the approach to chronicity.	Levels of NCM’s institutionalization: low, medium and advanced. ‑ Protocoled NCM’s functions. ‑ Legal security against other professional groups.	‑ There are no autonomous communities that have reached the maximum standard of implementation. ‑ There is a degree of erratic institutionalization of the NCM in Spain, despite its formal recognition.	‑ Recognition of the profile of the NCM involves knowing and understanding its differentiated character from the nursing profession. ‑ Policies are only kept on the institutional agendas if they respond to identified social problems. ‑ It is necessary to continue showing that the figure of the NCM is necessary and that it responds to the health needs of people, especially in complex chronicity.	STROBE: 16/22 SIGN: 4 D
Oh & Oh, 2017. South Korea.	Qualitative study Sample: 23 nurse case managers with three or more years of experience (in-depth interviews).	‑ To investigate issues related to the management of “Medical Aid” patients in the use of long-term hospitalization from the point of view of case managers.	‑ Obstacles wich hinder the progress of the work. ‑ Use of Korean health services. ‑ NCM’s limitations. ‑ Emotions felt by case managers.	‑ The target population has to be found for the development of case management. ‑ The Korean’s NCM does not have the authority to decide when a ‑ patient has recovered or has to be discharged. ‑ The most important task performed by the NCM is to defend the patients.	‑ This study shows the complexity facing the Korea’s NCM. ‑ Institutional support is needed for the implementation of NCM. ‑ Research by nurses is needed to enhance this field of care.	CASPe: 8/10 SIGN: 3 D
Joo & Huber, 2017.United States.	Scoping review.	To summarize the results of the case management.	‑ Weaknesses in nursing case management (lack of theoretical frameworks, lack of practice standards, role confusion) ‑ NCM’s strengh (continuous effort to develop theoretical frameworks, clear definitions of roles, self-assessment tools).	‑ Case management has proven to be effective in measuring biological and psychological indicator outcomes. ‑ Case management applied to nursing is effective and efficient, but the roles of the NCM need to be clearly defined.	‑ More evidence-based practice is needed in the context of the NCM, so more rigorous research is needed. ‑ Case management has great potential to improve the quality of care and introduce savings in health costs.	CASPe: 9/10 SIGN: 2++ B
Ozpancar, Cinar & Topcu, 2017. Turkey.	Randomized clinical trial. ‑ Sample: 60 randomly selected patients with hypertension who had no communication problems, with antihypertensive treatment for at least six months, of which 30 belonged to the study group and the other 30 to the control group.	‑ To determine the effect of the case management program on the treatment of patients with hypertension, as well as the adherence to antihypertensive medication.	‑ Adherence to the hypertensive medication scale. ‑ Assessment of the chronically ill patient at the first interview and six months later.	‑ There were no significant differences between the study group and the control group in the first interview. ‑ Adherence to antihypertensive medication and patient evaluation at 6 months is significantly higher in the study group compared to the control group.	Nurse-implemented case management resulted in a significant decrease in blood pressure in individuals with hypertension, improved adherence to treatment, and improved care for individuals with chronic diseases.	CASPe: 10/10 SIGN: 1++ A
Burguel et al., 2018. United States.	Qualitative and descriptive study.Sample: 80 patients with severe mental illness over 18 years old with at least one primary diagnosis, behavioral or emotional.	To promote a case management program to improve oral health.	‑ Assess oral health, dental treatment needs and quality of life related to oral health. ‑ Current dental goals ‑ Attend the first dental appointment. ‑ Self-care strategies to keep that first appointment.	‑ 72% of clients who participated in dental services also met with the case manager, with 87% keeping their first appointment. ‑ The initiative by the case managers included structural support. ‑ Client’s first appointment was an effective method for initiating and completing dental care.	‑ Case management program increases enthusiasm for dental care. ‑ The costs of dental health services continue to be a barrier to their use.	CASPe: 7/10 SIGN: 3 D
Davisson & Swanson, 2018. United States.	Mixed study (observations, interviews and coding were used).Sample: 6 adults over 65 years old in a rural hospital.	To describe how the nurse-led “Living Well” case management program impacts the care of chronically ill patients.	‑ Nurse’s knowledge ‑ Availability and value of the nurse ‑ Improvement of adhesion ‑ Participation of family and friends ‑ Self-management needs	‑ Nurse support was the main perceived benefit of the program. ‑ Nurse is an essential figure in supporting patient self-management. ‑ Nurses facilitate individual patient planning.	‑ The patients’ perception was positive and they highlighted the value of nurse-patient interactions. ‑ Need for programs of this nature that incorporate the role of the nurse manager and align with the values of the chronic care model.	CASPe: 8/10 SIGN: 3 D
Morales et al., 2019. Spain.	Longitudinal cohort follow-up study.-Sample: 835 patients.	To describe the characteristics of case management: interventions, service utilization, readmissions, pressure ulcers, falls	‑ Profile of patients included in the program ‑ Profile of caregivers. ‑ Clinical characteristics of the patients. ‑ NCM’s interventions.	‑ The most frequently performed interventions integral evaluation (87.2%), support to the PHCgiver (65.3%), counseling (45.9%), fall prevention (42.4%), medication management (41.2%) or active listening (35.8%). ‑ Hospital admissions corresponded to 38.1% of the total patients; 17.1% had a fall; 14.1% pressure injury and 5.5% drug-related problems.	‑ Patients who receive case management present a high complexity: comorbidities, dependence… ‑ Patients with multi-morbidity have high mortality, increased risk of falls or pressure ulcers… ‑ There is a high accessibility to case management services. ‑ It is necessary to analyze the levels of variability in the interventions towards patient isogroups (grouped by age, sex, functionality and clinical profile…).	CASPe: 11/11 SIGN: 2++ B
Sánchez et al., 2019. Spain.	Scoping review.	Implementing the role of the Advanced Practice Nurse and measuring its impact on clinical practice and patient benefit	‑ Quality of life. ‑ Profitability. ‑ Health results. ‑ Satisfaction. ‑ Accessibility.	‑ The generalization of the NCM’s implementation is the starting point for the development of advanced nursing practice in Spain, progressively implementing advanced practice in Spain. ‑ This implementation would have a positive effect on patients.	‑ APN can more effectively follow up on chronic patients. ‑ APN can increase the quality of life, improve patient empowerment, improve patient satisfaction, and reduce costs and waiting lists. ‑ NCM’s implementation should be prioritized for chronic and highly dependent patients.	CASPe: 9/10 SING: 2++ B
Scherlowski et al., 2020. Spain and Brasil.	Mixed study: bibliographic review of papers and publications on the nurse case manager and qualitative approach of interviews to 7 teaching nurses.	‑ To analyze NCM’s role, discuss aspects of its work and its possible implementation in the Brazilian Health System.	‑ Understanding the health-disease process by the NCM. ‑ Ability to respond to health needs in a low-cost context.	‑ The correct development of the NCM provides better health outcomes for patients. ‑ It is necessary that other professionals involved in the health of the patients know the NCM’s figure. ‑ NCM must have a concrete profile. ‑ NCM can discover unknown community resources.	‑ The Spanish experience can serve as a support to develop in the Brazilian Health System, with a figure capable of responding to the complex needs of people.	Bibliographic review CASPe: 9/10 Qualitative study CASPe: 8/10 SIGN: 3 D

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
