# Peer review of "Results of Nurse Case Management in Primary Heath Care: Bibliographic Review"

_ijerph, 2020, doi:10.3390/ijerph17249541_

Round 1

Reviewer 1 Report

I recommend for a future publication to carry out the review by extending the years of inclusion.

The main strength of the article is the chosen topic, which is very interesting for the holy assistance in the community. The main devility is the selection of the period of only five years, it should increase.

Author Response

I recommend for a future publication to carry out the review by extending the years of inclusion.

The main strength of the article is the chosen topic, which is very interesting for the holy assistance in the community. The main devility is the selection of the period of only five years, it should increase.

Thank you, we appreciate your comments. The study was done in the last 5 years to be as recent as possible and only with new information. We will consider your suggestion in further revisions.

Reviewer 2 Report

The article is well designed and well written. We have only minor "formal" comments:

1) In table 2 it should be more adequate to establish the order according to the date of publication.

2) In references through the text it should be more adequate to cite the numbers in many occassion. For example, in the Results change "four systematic reviews (Reilly et al.[21], Joo and Liu [22], Joo and Huber [23] and Stokes et al. [24]....." to "four systematic reviews [21-24], one of them including a meta-analysis" [24]....... 

Perhaps it could benefit from using other known databases besides PubMed, such as EMBASE, MEDLINE, and WEB OF SCIENCE (core collection). The use of these databases could increase the strengths of the document.

Author Response

The article is well designed and well written. We have only minor "formal" comments:

1) In table 2 it should be more adequate to establish the order according to the date of publication.

Thank you, we appreciate your suggestion. We have proceeded to modify it to make it more logical.

2) In references through the text it should be more adequate to cite the numbers in many occassion. For example, in the Results change "four systematic reviews (Reilly et al.[21], Joo and Liu [22], Joo and Huber [23] and Stokes et al. [24]....." to "four systematic reviews [21-24], one of them including a meta-analysis" [24].......

Thanks again, we proceed to make the change as you propose.

Perhaps it could benefit from using other known databases besides PubMed, such as EMBASE, MEDLINE, and WEB OF SCIENCE (core collection). The use of these databases could increase the strengths of the document.

Thank you. We appreciate your feedback. The bases included in the study (Pubmed, Scopus, Scielo and Dialnet) were carefully selected.

Pubmed is medline's search engine, so we consider that it is adequately represented. Also WoS contains a large number of journals indexed in Pubmed, and many of them are also indexed in SCOPUS.

On the other hand, with the other two indexes (Scielo and Dialnet) we tried to obtain studies written in Spanish language, which could cause them not to be located in Pubmed or Scopus.

However, we are aware of this limitation and therefore we have catalogued this review as a bibliographic review, and not as a systematic review.

Reviewer 3 Report

The work is very interesting, the implementation of the NCM is important. However, I think it is appropriate to make some observations to the authors:
Methodology:
Although the methodology does not follow a pre-established review structure, it is possible to trace the sequence. Nevertheless, I suggest to improve the next aspects:
1. The search strategy table is not complete. It is missing the dates of the searches and the date of the last search. Authors should clearly define in the table the filters or limits of the search, not only the period of selection of articles.
2. The search strategy does not include all the articles in the bibliography. With the exception of 4 methodological articles, the rest must be included either in the same table or in a manual search table and justify their use in the work.
3. It is advisable to separate the inclusion criteria, which must correspond to the objectives of the research, from the filters or limits of the search that do not respond to specifically scientific criteria.
4. With regard to the exclusion criteria, these are not the antonyms of the inclusion criteria. The authors must clearly define the criteria with which they do not select articles that previously respond to the objectives of the work and yet decide not to include them.
5. Do the research variables refer to the data list? Even if a structured model of review is not followed, they should be identified as such.

Outcomes:
1. Why is the quality and evidence synthesis of all articles not analysed before inclusion in the study and why do they not appear in the search strategy? Articles included in the manual or reference search should be included in the search strategy table or in a manual search table and justify why they were used.
2. All articles in the study should respond to the objectives and therefore be analysed with instruments that guarantee their inclusion in a literature review, even if they do not correspond to a defined model. That is, they should be included in Table 2

Discussion:
1. The authors should show in the discussion the levels of evidence found in order to determine the main outcome and the answer to the research question based on the criteria of the best available evidence.This helps to elucidate possible biases or limitations.

Bibliography:
2. The bibliography should be revised and unified in its format, either the simplified Vancouver format or the complete one. It is not necessary to describe [online] when the web reference appears. The date of consultation should be included in all web citations. The reference information must be unified. If DOI is used, it must be integrated in all citations, without ignoring the web address. If you use the publication search filter in the years 2015-2020, why are you using articles from the years 2012 and 2013? This must be justified in the methodology, and only one specificity is mentioned for the EMBASE database.

Author Response

The work is very interesting, the implementation of the NCM is important.

Thank you for your comment, We appreciate that You find it interesting.

However, I think it is appropriate to make some observations to the authors:
Methodology:
Although the methodology does not follow a pre-established review structure, it is possible to trace the sequence. Nevertheless, I suggest to improve the next aspects:
1. The search strategy table is not complete. It is missing the dates of the searches and the date of the last search. Authors should clearly define in the table the filters or limits of the search, not only the period of selection of articles.

Since we do not have access to the specific dates of each search, but we do have access to the last search, it has been added in the text:

"The last search was conducted on April 30, 2020"

2. The search strategy does not include all the articles in the bibliography. With the exception of 4 methodological articles, the rest must be included either in the same table or in a manual search table and justify their use in the work.

Thank you for your comment, We appreciate it very much. In the model search strategy table that We intend to write, all the articles that have been used in the results section (refs 10, 17-31) are collected, which at the same time are the ones used in the discussion.

The rest of the articles in the references are not all derived from the systematic search, but from free searches, and have been used exclusively to generate the introduction and the theoretical framework, which is why We justify not including them in the table of results.

3. It is advisable to separate the inclusion criteria, which must correspond to the objectives of the research, from the filters or limits of the search that do not respond to specifically scientific criteria.

Thank You very much for the comment, it has clarified for us further the differentiation between the criterion of inclusion and exclusion. The temporary filter has been removed from the inclusion criteria, and has been moved to the exclusion criteria, thus remaining:

"Recovered articles before 2015 (in Scopus the filter of years 2016-2020 was used, due to the limitations of the database)".

4. With regard to the exclusion criteria, these are not the antonyms of the inclusion criteria. The authors must clearly define the criteria with which they do not select articles that previously respond to the objectives of the work and yet decide not to include them.

Thank You, We have proceeded to modify it as well, eliminating the exclusion criterion that spoke of the objectives, and including the following exclusion criterion:

"Articles describing the role of NCM without studying the results of their interventions".

5. Do the research variables refer to the data list? Even if a structured model of review is not followed, they should be identified as such.

Thank you!

Yes, the research variables refer to how the data are grouped and presented in the results section. You can check how the results section has been structured in 2 sections: efficacy and efficiency.

Outcomes:
1. Why is the quality and evidence synthesis of all articles not analysed before inclusion in the study and why do they not appear in the search strategy? Articles included in the manual or reference search should be included in the search strategy table or in a manual search table and justify why they were used.

Thank You very much, it's an interesting commentary. Since the selected documents were analyzed from critical reading processes with quality evaluation guides, the inclusion and exclusion criteria were previously applied so that the guides were applied exclusively to the documents that provided the intended information.

As for the inclusion commentary in the strategy table, We have proceeded to incorporate it, also setting the dates of the 3 articles obtained in manual search, and in the table footer the reason for its selection has been explained.

2. All articles in the study should respond to the objectives and therefore be analysed with instruments that guarantee their inclusion in a literature review, even if they do not correspond to a defined model. That is, they should be included in Table 2

All the articles that provided the information that responded to the objectives were incorporated into the results table.  This has not been the case with the documents in the introduction (with the exception of reference 10), which do not derive from the systematic search.

Discussion:
1. The authors should show in the discussion the levels of evidence found in order to determine the main outcome and the answer to the research question based on the criteria of the best available evidence.This helps to elucidate possible biases or limitations.

Thank You very much for this suggestion, it is very interesting and has been incorporated into the discussion.

Bibliography:
2. The bibliography should be revised and unified in its format, either the simplified Vancouver format or the complete one. It is not necessary to describe [online] when the web reference appears. The date of consultation should be included in all web citations. The reference information must be unified. If DOI is used, it must be integrated in all citations, without ignoring the web address. If you use the publication search filter in the years 2015-2020, why are you using articles from the years 2012 and 2013? This must be justified in the methodology, and only one specificity is mentioned for the EMBASE database.

Thank you.The style used is Chicago 2, the one proposed by the magazine, slightly different from vancouver. We have proceeded to carefully review the bibliographic style, eliminating the online allusions as you propose.

As we have already explained in a previous commentary, the results of the search do not include the articles excluded from it, some of which have been used for the introduction, background, theoretical framework and justification of the problem, but not to define the conclusions or evidence.

Reviewer 4 Report

Authors of the manuscript presented a short review about Nurse Case Management (NCM), which may be an interest to the readers of IJERPH.

Major point:

  • Manuscript is a review about NCM from 2015-2020, however the article is presented like a meta-analysis, and not following the structure of a “traditional” review article. Why did authors approach this form of presentation? Reviewer suggests to re-organize the article to be more similar to those of traditional reviews.

Minor points:

  • Abbreviations should be checked (i.e. NCM is three times CMN, PC/PHC should be also corrected)
  • Table 1: please include in a new row the additional +3 articles as well.
  • Methods: Text seems to be way too fragmented, merging paragraphs that are connected is recommended.
  • Line 33: use nursing instead of this nurse
  • Line 119: “Bibliographic review” was “carried…”
  • Line 199: professional – patient relationship
  • Line 138-139: citation or Author should be inserted

Author Response

Authors of the manuscript presented a short review about Nurse Case Management (NCM), which may be an interest to the readers of IJERPH.

Thank you for your comment, we appreciate that you find it interesting.

Major point:

  • Manuscript is a review about NCM from 2015-2020, however the article is presented like a meta-analysis, and not following the structure of a “traditional” review article. Why did authors approach this form of presentation? Reviewer suggests to re-organize the article to be more similar to those of traditional reviews.

Thank You for your comment, We appreciate your advice. We have proceeded to follow a strict methodology of systematic review, incorporating quality and evidence evaluations. With this, we have only tried to provide rigour to the process of organization and structure of the article.

Evidently, We have not carried out any type of meta-analysis with the recovered articles or with the data obtained in the review, among other reasons because it was neither the objective nor the heterogeneity of the designs allows it. Therefore, without being a meta-analysis, we have tried to ensure that the structure and organization of the manuscript is, as we have said, that of a systematic review.

Minor points:

  • Abbreviations should be checked (i.e. NCM is three times CMN, PC/PHC should be also corrected)

Thank You, we have proceeded to review and correct it for errors. 

  • Table 1: please include in a new row the additional +3 articles as well.

Thank You for the comment. The information has been added to table 1.

  • Methods: Text seems to be way too fragmented, merging paragraphs that are connected is recommended.

Thank You, We have proceeded to reunify some related paragraphs, which are marked in red in the text as they are joined.

  • Line 33: use nursing instead of this nurse

Thank You, We have modified nurse, following your advice.

  • Line 119: “Bibliographic review” was “carried…”

Thank You, We have modified it.

  • Line 199: professional – patient relationship

Thank You, We have modified it.

  • Line 138-139: citation or Author should be inserted

Thank You, We have modified it.

Round 2

Reviewer 4 Report

Authors answered all my concerns.

During proofreading, within the abstract, please correct the English of the following sentence: "Inclusion criteria: articles written in the last 5 years, 32 which analyze how this nursing rol influences the care and health of patients."